# Spatiotemporal Big Data for PM_2.5_ Exposure and Health Risk Assessment during COVID-19

**DOI:** 10.3390/ijerph17207664

**Published:** 2020-10-21

**Authors:** Hongbin He, Yonglin Shen, Changmin Jiang, Tianqi Li, Mingqiang Guo, Ling Yao

**Affiliations:** 1School of Geography and Information Engineering, China University of Geosciences, Wuhan 430074, China; 20161004430@cug.edu.cn (H.H.); 20141000446@cug.edu.cn (C.J.); ltq@cug.edu.cn (T.L.); 2Institute of International Rivers and Eco-security, Yunnan University, Kunming 650500, China; 3State Key Laboratory of Resources and Environmental Information System, Institute of Geographic Sciences and Natural Resources Research, Chinese Academy of Sciences, Beijing 100101, China; yaoling@lreis.ac.cn

**Keywords:** spatiotemporal big data, empirical orthogonal function (EOF), geographic weighted regression (GWR), population distribution, COVID-19

## Abstract

The coronavirus disease 2019 (COVID-19) first identified at the end of 2019, significantly impacts the regional environment and human health. This study assesses PM_2.5_ exposure and health risk during COVID-19, and its driving factors have been analyzed using spatiotemporal big data, including Tencent location-based services (LBS) data, place of interest (POI), and PM_2.5_ site monitoring data. Specifically, the empirical orthogonal function (EOF) is utilized to analyze the spatiotemporal variation of PM_2.5_ concentration firstly. Then, population exposure and health risks of PM_2.5_ during the COVID-19 epidemic have been assessed based on LBS data. To further understand the driving factors of PM_2.5_ pollution, the relationship between PM_2.5_ concentration and POI data has been quantitatively analyzed using geographically weighted regression (GWR). The results show the time series coefficients of monthly PM_2.5_ concentrations distributed with a U-shape, i.e., with a decrease followed by an increase from January to December. In terms of spatial distribution, the PM_2.5_ concentration shows a noteworthy decline over the Central and North China. The LBS-based population density distribution indicates that the health risk of PM_2.5_ in the west is significantly lower than that in the Middle East. Urban gross domestic product (GDP) and urban green area are negatively correlated with PM_2.5_; while, road area, urban taxis, urban buses, and urban factories are positive. Among them, the number of urban factories contributes the most to PM_2.5_ pollution. In terms of reducing the health risks and PM_2.5_ pollution, several pointed suggestions to improve the status has been proposed.

## 1. Introduction

Coronavirus disease 2019 (COVID-19) is a lung disease caused by a novel coronavirus first detected in late 2019, which has a significant impact on the regional environment [1,2,3] and human health [4,5]. Many scholars discussed the clinical manifestations [6] of COVID-19 and the risk factors of death and its detailed clinical course [7,8], providing an important basis for the rapid and accurate diagnosis of COVID-19 patients. Airborne pollutants with diameters less than or equal to 2.5 microns are known as PM_2.5_ or fine particles. Due to the characteristics of small diameter, large-area coverage, strong activity, easy bonding with toxic substances, PM_2.5_ can be suspended in the atmosphere for a long time, severely impacting on the regional environment and Earth’s biological cycle [9]. Furthermore, they are prone to provide carriers for toxic substances, e.g., absorbing harmful gases such as polycyclic aromatic hydrocarbons (PAHs) from industrial exhaust gas and polluted microorganisms, which then enter the human body through breathing, causing harm to the immune system, respiratory tract, cardiovascular and cerebrovascular system, and nervous center system of the human body and causing a series of diseases. With the prevalence of the internet of things (IoT), artificial intelligence (AI) and cloud computing, the spatiotemporal big data, e.g., location-based service (LBS) and place of interest (POI) data, that derived from navigation and positioning, location and trajectory, online car-hailing order, social media network, and macro national economic, are increasing rapidly. The rational and effective use of spatiotemporal big data will play a positive role in understanding influence factors of PM_2.5_ concentration, and quantitatively assessing the population exposure and health risks of PM_2.5_ during COVID-19.

Compared the state-of-the-art studies on PM_2.5_ exposure and health risks, most studies used census-based population data, which ignores the spatial heterogeneity of population, and indirectly leads to lower accuracy. With the rise of the IoT, more and more portable mobile devices are adding LBS, a service that records real-time data about people’s activities in space and time. Compared to traditional demographics-based or global positioning system (GPS)-based location recording data, integrated LBS and PM_2.5_ data can improve the assessment of the population exposure and health risks because it can be modeled with high precision by establishing a relationship between PM_2.5_ concentrations and population density. The spatial and temporal distributions of PM_2.5_ have been extensively studied using geostatistical methods [10,11], as well as the pattern of spatial distribution of PM_2.5_ was analyzed. For China, most studies focused on the Jing-Jin-Ji urban agglomeration [12,13], the Yangtze River economic zones [14,15,16], and the Pearl River Delta region [17], Cheng-Yu urban agglomeration [18]. Studies at the national scale are also increasingly being undertaken. Li et al. [19] used statistical methods and geographic information system (GIS) technology to systematically analyze the PM_2.5_ concentration to discover the spatial and temporal distribution patterns of air pollution in the 161 cities of mainland China in 2014. In addition, Zhang et al. [20] found that PM_2.5_ is associated with an increase in disease morbidity and mortality by examining indicators of the biological effects of PM_2.5_. Fu et al. [21] established an empirical model of human health risk factors caused by excessive PM_2.5_ concentration, and studied the impact of the combination of population spatial distribution and PM_2.5_ spatial distribution on population health. Song et al. [22] used spatiotemporal big data (micro-blog location data) to study the Jing-Jin-Ji urban agglomeration of PM_2.5_ for dynamic exposure and health risk assessment. However, current studies rarely consider the mobility of the population when assessing the PM_2.5_ exposure and health risk, especially in the COVID-19 epidemic.

In this study, a comprehensive assessment of the population exposure, health risks, and driving factors of PM_2.5_ nationwide during the COVID-19 epidemic has been conducted by using high-precision spatiotemporal data. To be specific, the empirical orthogonal function (EOF) is used to analyze the patterns of PM_2.5_ concentrations in China in both the temporal and spatial dimensions. Then, high-precision LBS data are used to explore the potential relationship between Chinese population and PM_2.5_ exposure during the COVID-19 epidemic, and to quantitatively assess the corresponding health risk. In order to quantify the driving factors of PM_2.5_ that impact health risk, this study uses geographically weighted regression (GWR) method to quantify the relationship between POI and PM_2.5_. This study expects to provide comprehensive reference and decision making for better protecting the ecological environment.

## 2. Materials and Methods 

### 2.1. Study Area and Datasets

As a developing country, China is facing the twofold challenges of economic development and environmental protection. China’s economy is sustained growing, with a gross domestic product (GDP) of 990,865 million yuan in 2019, an increase of 67.1% compared with that of 2013. Meanwhile, energy consumption reached 4.86 billion tons of standard coal in 2019, a significant increase over total energy consumption in 2013. Environmental protection is a basic national policy and an important strategy for achieving sustainable development, and is an integral part of comprehensive pollution control and environmental protection planning. According to the China Air Quality Improvement Report. (2013–2018) [23] published by the Ministry of Ecology in 2019, overall air quality in China improved from 2013 to 2018, and the emissions of major air pollutants decreased significantly. The average concentration of PM_2.5_ in 74 cities using ambient air quality standards fell by 42%. China has also actively formulated a national climate change strategy to promote environmental improvement by reducing carbon emissions.

The data involved in this study mainly include PM_2.5_ monitoring station data, POI data, national secondary industry output value data, a digital elevation model (DEM), LBS data, and urban population data (Figure 1). 

(1)PM_2.5_ concentration was obtained from the National Real-time Urban Air Quality Release System of the China environmental monitoring general station [24] (national air quality monitoring stations were in only 190 cities in 2014, and since then all cities have been covered). Data from 1670 stations were acquired until 2020. To present the spatiotemporal distribution of PM_2.5_, a GIS-based spatial interpolation is applied to convert to 1 × 1 km grid.(2)POI data, were derived from the National Bureau of Statistics [25] and the local bureau of statistics, mainly including six factors that affect the distribution of PM_2.5_ concentration, including urban GDP, green space, road area, the number of urban taxis, the number of urban buses, and the number of urban factories. Urban GDP includes primary, secondary, and tertiary industry GDP, in units of 10,000 yuan. The urban green area contains public, residential, unit affiliated, protective, production, road green, and scenic forest areas, and the unit is hectares. The road area includes urban main roads, secondary, and branch roads, with a unit of 10,000 m^2^. The number of urban taxis and buses is the total number of vehicles in operation at the end of each year. The number of urban factories contains capital-intensive industrial, labor-intensive, resource-intensive, and knowledge-technology-intensive factories. The time period of all POI data was from 2014 to 2017.(3)LBS data came from Tencent location big data [26], which use positioning technology to obtain a position in the whole scene. Relying on the full coverage and high accuracy of LBS data, to study the impact of the COVID-19 epidemic on population mobility, this study selected four main time nodes because of the limited permission to access the Tencent’s services, i.e., 25 December 2020, 11 January 2020, 7 March 2020, and 3 April 2020. On 25 December 2019 people in China were producing and living normally. The turning point of the COVID-19 outbreak was 11 January 2020. Since this period, the number of patients infected with COVID-19 has continued to increase. It is also the time node when the Spring Festival travel season began. The peak number of COVID-19 infections was on 7 March 2020. On 3 April 2020 the COVID-19 epidemic in China was basically under control, and the life of the people returned to normal. The temporal resolution of LBS data is the hour, and the data accumulatively obtained 297,535,280 positioning times of 4,194,304 positioning points. The study uses the grid mapping method of the geographic information system to map the positioning data to a 1×1 km grid, and the number of positioning of all the positioning points in the same grid is summed to obtain the population of the grid.(4)Auxiliary data contained 2017 national secondary industry output value data and 2019 national urban population data from the National Bureau of Statistics [25], and DEM data were from the Resource and Environment Science and Data Center [27].

### 2.2. Empirical Orthogonal Function (EOF)

EOF was first proposed by the statistician Pearson in 1902 [28], and is known as the spatiotemporal decomposition in geological applications [29,30,31,32,33,34,35]. The absolute value of the time series coefficient represents the degree of air pollution, the closer it comes to zero, better the air quality is. This study firstly calculated the average PM_2.5_ concentration $\bar{P_{ij}}$ of 1657 stations in 367 prefecture-level cities in 31 provinces of China (excluding Hong Kong, Macao, and Taiwan) from 2014 to 2019, then estimates the PM_2.5_ concentrations anomalies $P_{mn}$ and the covariance matrix $A_{jk}$,
(1)$$P_{mn}=\left(P_{ij}-\bar{P_{ij}}\right)$$
(2)$$A_{mk}=\frac{1}{m}\cdot P_{mn}\cdot {\left(P_{nk}\right)}^{T}$$
where i,m represents the year, i=m=1,2,…,6; and j,k,n is the station at each city, j=k=n=1,2,…, 1657; $A_{mk}$ is a real symmetric, positive definite square matrix of order n. The Jacobi method was used to solve the eigenvalues $\lambda_{1}\geq \lambda_{2}\geq \lambda_{3}\geq \cdots \geq \lambda_{k}\geq \cdots \geq \lambda_{n}$ and eigenvectors of the covariance matrix $A_{jk}$. The eigenvalues and corresponding eigenvectors can be formulated by,
(3)$$A_{nn}\cdot X_{k}=\lambda_{k}\cdot X_{k}$$
where $X_{nn}$ is the natural orthogonal function of the original field $P_{mn}$, and its time series coefficient can be expressed as,
(4)$$T_{mn}=P_{mn}\cdot X_{nn}=\left[\begin{array}{cccc} T_{11} & T_{12} & \ldots & T_{1n}\\ T_{21} & T_{22} & \ldots & T_{2n}\\ \ldots & \ldots & \ldots & \ldots \\ T_{m1} & T_{m2} & \ldots & T_{mn} \end{array}\right]$$
where $T_{mn}$ and $P_{mn}$ are matrices with the same order. The original field can be represented as a linear combination of natural orthogonal functions,
(5)$$P_{mn}\left(t,s\right)=T_{mn}\left(t\right)\cdot X_{nn}\left(s\right)$$
where s represents the space point and t represents the observation time. In general, the most important information in the original field can be fully reflected by the first few eigenvectors and time series coefficients.

### 2.3. Particulate Matter (PM_2.5_) Exposure Assessment

LBS data are adopted to conduct PM_2.5_ exposure assessment and health risk analysis, since LBS data are with a daily temporal resolution, which can accurately reflect the population mobility [36]. In this study, the LBS data are employed as an indicator to quantify the spatial and temporal patterns of population distribution. Due to the differences of socio-economic development and mobile internet penetration among different cities, the population dynamic distribution of each city is estimated separately. Since the daily total location data contains duplicate data, in order to eliminate the impact of duplicate data, this study redistributes the total population data of each city based on the hourly LBS data location data (Equation (6)), and generates the density map of LBS data by aggregating all the geotagged records of each grid.
(6)$$Pop_{pq}=\frac{P_{pq}}{\sum_{p=1}^{z}P_{pq}}\cdot Tol_{p}$$
where $p_{pq}$ is the positioning quantity of Tencent’s position data in the p grid; z is the total number of pixels in the city; $Tol_{p}$ is the total population of the city. For each city, the PM_2.5_ exposures were assessed via a pixel-based method (Equation (7)) [37], which can effectively reduce the potential zoning effect of the modifiable areal unit problem (MAUP) [38].
(7)$$E_{PM}=\frac{Pop_{p}\times PM_{p}}{\sum_{p=1}^{n}Pop_{p}}$$
where $PM_{p}$ stands for PM_2.5_ concentration; $Pop_{p}$ is the data redistributed based on LBS data. $E_{PM}$ represents a population-weighted assessment of PM_2.5_ exposure.

### 2.4. Health Risk Assessment of PM_2.5_ during COVID-19 Outbreaks

In this study, the health risk of PM_2.5_ was evaluated using the long-term risk assessment value recommended by the World Health Organization (WHO). With reference to the assessment method of the WHO, the health risk of residents was assessed using the PM_2.5_ concentration obtained and the current status of population distribution.
(8)$$E=\beta \cdot \left(c-c_{0}\right)\cdot E_{0}\cdot P_{p}$$
where E represents the potential risk of disease of residents under the current PM_2.5_ concentration; β is the proportion of the mortality caused by the change of PM_2.5_. In this study, the mortality rate of PM_2.5_ decreased by 10 μg/m^3^ by 6% [39]; c is the actual PM_2.5_ concentration; $c_{0}$ is the referenced PM_2.5_ concentration; $E_{0}$ is the health effect of residents under the referenced concentration. The mortality rate in the population census (7.14%) was selected in this study; and $P_{p}$ is the population at the current location.

### 2.5. Geographic Weighted Regression (GWR)

General linear and non-linear regression were widely used in previous studies to analyze the relation between two groups of variables, However, taking consideration of heterogeneity of POI data, an advanced method of GWR is adopted in this study to eliminate spatial heterogeneity and improve the fitting results. Fotheringham et al. [40], based on the thinking behind the local smooth GWR model, is put forward [41,42,43]. The GWR model can be formulated by,
(9)$$y_{p}=\beta_{0}\left(u_{p},\;v_{p}\right)+\beta_{1}\left(u_{p},\;v_{p}\right)\cdot x_{p1}+\cdots +\beta_{q}\left(u_{p},\;v_{p}\right)\cdot x_{pq}+\varepsilon_{s}$$
where $\left(u_{p},\;v_{p}\right)$ represents the spatial position p, (p=1,2,…,367). The spatial regression coefficient of $\beta_{0}$, $\beta_{1}$, …$,\beta_{q}$ is a function of position space coordinates which indicates the degree that spatial independent variables affected dependent variables. $\varepsilon_{s}$ is a deviation, it represents the degree of deviation of two variables. $y_{p}$ is PM_2.5_ concentration. $x_{pq}$ is POI data.

## 3. Results and Discussion

This study focuses on the following four parts: (1) through spatial autocorrelation analysis and EOF decomposition, discussing the temporal and spatial characteristics of PM_2.5_ concentration; (2) based on the spatiotemporal distribution of PM_2.5_, calculating the population exposure of PM_2.5_ based on Tencent LBS data during the COVID-19 epidemic; (3) adopting the long-term health assessment method of air quality recommended by the World Health Organization (WHO) to assess the health risk of PM_2.5_ during the COVID-19 epidemic; (4) the relationship between PM_2.5_ and its driving factors were quantitatively analyzed using the GWR method to address the health threat posed by PM_2.5_ to human beings, and relevant measures are proposed to address these driving factors. 

### 3.1. EOF Analysis of Monthly PM_2.5_ Concentration

This study used the EOF spatiotemporal decomposition of PM_2.5_ station observations from 367 cities during 2014–2019 to reveal the spatial and temporal patterns of PM_2.5_. Because the first feature vector has a cumulative contribution of variance greater than 70% (Figure 2), it was chosen to characterize the spatial pattern of PM_2.5_ concentration.

The time series coefficient reflects the variations of PM_2.5_ concentration over time. Figure 3 shows the change in time series coefficients corresponding to the PM_2.5_ eigenvector for each month during the six-year period of 2014–2019. The time series coefficients of PM_2.5_ concentration have the same changing trend, and the difference between months is significant, with the peak intervals located in the month of January from 2015–2019. The valley value of the time series coefficient of PM_2.5_ concentration is located in August 2014, August 2015, August 2016, August 2017, September 2018 and August 2019. Overall, the time series coefficients for each year show a U-shaped distribution characteristic of increase after an initial decrease. In China, precipitation is scarce in January, February and December each year, and the phenomenon of “temperature inversion” is prominent, which seriously hinders the horizontal transportation and vertical diffusion of air, and easily causes pollutants to gather in the surface layer, making PM_2.5_ pollution more serious [44]. Meanwhile, the temperature in January, February, and December is low, and people use more electricity to keep warm. Insufficient combustion of fossil fuels by facilities such as thermal power plants and coal furnaces causes a substantial increase in PM_2.5_ concentrations [45]. In addition, the large amount of straw burning also contributes to a high concentration of PM_2.5_ [46]. According the variation of time series coefficient from 2014–2019, values of 2018 and 2019 are relatively low, indicating that PM_2.5_ concentration declined significantly from 2018.

The spatial distribution of the EOF first feature vector reflects the overall spatial distribution characteristics of PM_2.5_ concentrations on the monthly scale. From Figure 4, it can be seen that the high PM_2.5_ concentration areas in China in 2014 were mainly concentrated in the Jing-Jin-Ji urban agglomeration and Heilongjiang Province, and the high PM_2.5_ concentration areas from 2015 to 2019 were concentrated in Hebei Province, Henan Province, and Hubei Province and the spatial distribution of PM_2.5_ concentration sprawling from the area to its surroundings. As seen in Figure 1a, the Jing-Jin-Ji urban agglomeration, Henan, and Shandong Province are located in the lower elevation plain, and surrounded by mountains, which are not conducive to the atmospheric transmission of PM_2.5_. In addition, the irrational industrial structure within these regions leads to the emission of endogenous pollutants [47]. Since 2015, the Xinjiang Autonomous Region (XAR) started to appear as the region of second highest PM_2.5_ concentration, with peaks in Urumqi and Kashgar in 2019. On the one hand, because XAR is far from the ocean, precipitation is scarce, and it has China’s largest desert, the Taklamakan Desert, whose wind fields basically dominate in the low precipitation and drought of XAR [48]. On the other hand, the population of XAR is growing rapidly, and at a rate higher than the national average [49]. Increasing winter heating facilities and population growth has had a serious impact on the fragile environment of the XAR, which causes serious PM_2.5_ pollution. In addition, there are many mines around Urumqi. With the development of Midong District, high-pollution industries such as petrochemicals, chlor-alkali chemicals, coal, electricity and coal chemicals have gathered here [50], and large freight vehicles have a large traffic flow, which is affected by secondary particles and coal. The impact of smoke and vehicle emissions is greater, resulting in serious PM_2.5_ pollution around Urumqi. PM_2.5_ concentrations reached troughs in southwest Yunnan Province, southwest Sichuan Province, Tibet Autonomous Region, and southeast coastal areas. On the one hand, the underdeveloped economy (Figure 1c) and low industrial output (Figure 1d) in these regions are the predominant reasons. On the other hand, the terrain of these areas is rugged (Figure 1a), and the primary industry, tertiary industry service industry and tourism are mainly developed, which have less impact on PM_2.5_, so the level of PM_2.5_ pollution in these areas is low.

### 3.2. PM_2.5_ Population Exposure Assessment

In order to analyze the changes in PM_2.5_ population exposure during the COVID-9 epidemic. Based on the spatiotemporal distribution of PM_2.5_, this research selects LBS big data during the COVID-19 epidemic, and uses a GIS-based method to allocate the acquired population data to a grid of 1 × 1 km. On this basis, we have calculated a population density distribution map (Figure 5). 

As shown in Figure 5, the maximum value areas of national population density in December 2019 are distributed in the North China Plain, the Yangtze River Delta urban agglomeration, the Pearl River Delta urban agglomeration, the Chengdu-Chongqing urban agglomeration and Central China. The minimum areas are mainly distributed in the western and northwestern regions of China. In January 2020, the maximum value areas are concentrated in the North China Plain, the Yangtze River Delta urban agglomeration, the Pearl River Delta urban agglomeration, the Chengdu-Chongqing urban agglomeration, and Central China. January 2020 compared to December 2019, when the range of maximum value areas is decreasing, and population density shows a trend of spreading from high value areas to the low. Statistics by the State Railway Administration [51] and the Civil Aviation Administration of China (CAAC) [52] show that the number of transported passengers was 321,862,000 in January 2020. The degree of population mobility is large, and the trend of mobility is the movement of people from first-tier and new first-tier cities such as Beijing, Shanghai, Guangzhou, Shenzhen, and Wuhan to surrounding smaller cities. 

In March 2020, the scope of the region of maximum population density compared to January 2020. The region of greatest values is concentrated in the Yangtze River Delta, Pearl River Delta, and Chengdu-Chongqing urban agglomeration. As a result of the COVID-19 epidemic, the number of COVID-19 infected patients in the country peaked in March 2020, while there were no large number of new suspected infections, access to various community units in the country was strictly controlled, and road, railway and air transport authorities also took corresponding measures to restrict population movement. Enterprises in some cities gradually resumed work and production in March, but employees had to be quarantined for 14 days for observation before resuming work and production, resulting in lower urban population density in first-tier and new first-tier cities in China. In April 2020, the maximum population density areas of large cities were concentrated in the Jing-Jin-Ji urban agglomerations, the Yangtze River Delta, the Pearl River Delta, the Chengdu-Chongqing urban agglomeration, Wuhan and Changsha, with an increasing trend compared to March 2020, but there is still a gap compared to the population density in January 2020, and with the state’s push for enterprises to resume work and production, most of the enterprises have started production activities in April 2020, and the population density range has an increasing trend in the first-tier and new first-tier cities. In general, the more densely populated areas are mainly distributed in the middle and lower reaches of the Yangtze River Delta (Shanghai, Jiangsu, Zhejiang, Anhui, etc.), the Pearl River Delta (Guangzhou, Shenzhen, Hong Kong Special Administrative Region of China, Macao Special Administrative Region of China, etc.), the Jing-Jin-Ji urban agglomeration (Beijing, Tianjin, Hebei, etc.), and the Chengdu-Chongqing urban agglomeration (Chongqing, Chengdu, Mianyang, etc.). The total economic output and per capita GDP are relatively high in the region. The population distribution throughout China is characterized by a relatively high concentration of people in the eastern part of the country and a relatively sparse population in the western part.

At the end of 2019, a sudden outbreak of COVID-19 epidemic, bringing a heavy blow. From the comparison between Figure 5a,b, the population density of Hubei changed greatly from 25 December 2019 to 11 January 2020, and the change in Wuhan City was particularly noticeable. The official report by the Wuhan Municipal Railway Bureau on 11 January 2020, the stations under the jurisdiction of the Wuhan Railway Bureau sent 570,000 passengers, including about 100,000 student passengers. As can be seen from Figure 6c, the population density of Hubei has also declined significantly compared to January 2020, with the range of population density in Wuhan shrinking considerably. Due to the COVID-19 outbreak in Wuhan, the city of Wuhan has been blocked to management since 24 January 2020, and the degree of population movement has decreased, reducing the spatial range of the maximum value of population density in Wuhan. From Figure 5d, the spatial extent of population density in cities of Hubei, except Wuhan, tends to expand, accompanied by a gradual intensification of population mobility with the resumption of work in Hubei.

To further quantify the changes of the COVID-19 situation on PM_2.5_ exposure and the effectiveness of LBS data, data analysis on five typical population-intensive cities of China (i.e., Beijing, Shanghai, Guangzhou, Chengdu, and Wuhan) was performed, in which the daily PM_2.5_ ground monitoring station data [24], demographic data and LBS data were used. In Figure 6, green curves are daily PM_2.5_ exposure based on demographic data for the period of December 2019 to April 2020 (the duration of COVID-19); by contrast, blue curves are the reference of PM_2.5_ exposure using historical average data (December 2014 to April 2019); marks (red star) represent LBS-based PM_2.5_ exposure on four dates mentioned above.

The influence of the COVID-19 situation on air pollution can be concluded by the comparison of demographic-based PM_2.5_ exposure, i.e., the result in the duration of COVID-19 (green curve) and the result in the historical average (blue curve). Take Wuhan for example, as show in Figure 6e, we found that the PM_2.5_ exposure in Wuhan during the COVID-19 pandemic saw a significantly drop due to the community containment measures. Similarly, cities such as Beijing, Shanghai, Guangzhou, Chengdu, follow the same pattern. The effectiveness of LBS data can be concluded by the comparison of demographic-based PM_2.5_ exposure in the duration of COVID-19 (green curve) and LBS-based PM_2.5_ exposure on four dates (red star). Although we only obtained LBS data at four key time nodes because of access restrictions, the trend of LBS-based PM_2.5_ exposure (red star) is consistent with demographic-based PM_2.5_ exposure (green curve) in these cities. It demonstrates that LBS data on four key dates can characterize the variation of PM_2.5_ exposure well during the COVID-19 pandemic.

As shown in Figure 7, the areas with the highest exposure values on 25 December 2019 occurred in Henan, Hebei, Beijing, the northwestern part of Shandong, the Pearl River Delta, Chengdu, Chongqing, and the capital cities of the Northeast, the exposure values of other cities are relatively low. The areas with the largest exposure values on 11 January 2020 were mainly concentrated in Henan, Beijing, Tianjin, Hebei, Shanghai, Heilongjiang, and the Pearl River Delta. The area with the highest PM_2.5_ exposure on 7 March 2020 appeared around Shanghai. The maximum value on 3 April 2020 appeared in the Pearl River Delta, Wuhan and surrounding areas of Shanghai. According to the trend of the exposure values of PM_2.5_ from December 2019 to April 2020, it can be seen that the exposure values of large cities such as the Jing-Jin-Ji urban agglomeration, the Pearl River Delta, the Yangtze River Delta, and the Chengdu-Chongqing urban agglomeration have maintained at a relatively high level. From December 2019 to April 2020 in these regions, the exposure of PM_2.5_ gradually decreased. Because these areas were central cities for economic development, with a large population density and dense urban space, PM_2.5_ is difficult to diffuse, the values of exposure are large. From Figure 7a,b, the PM_2.5_ exposure in large cities in January 2020 is lower than that in December 2019. It is related to the Spring Festival travel. With the migration of the population, PM_2.5_ exposure in the Jing-Jin-Ji urban agglomeration relatively reduced. Combined with the spatial distribution of PM_2.5_, the reason for the decrease in the exposure of PM_2.5_ in March 2020 compared with January 2020 is related to the control measures issued by government in response to the COVID-19 epidemic. When the COVID-19 occurred, all provinces across the country (autonomous regions and municipalities directly under the central government) launched a first-level emergency response to major public health emergencies. All communities, streets, villages, and towns across the country have implemented community isolation through grid management, and the flow of people has been greatly reduced. In addition, the provinces announced that in addition to guaranteeing the operation of public utilities (e.g., water supply, gas supply, power supply, communication and other industries), epidemic prevention and control (e.g., medical equipment, medicines, protective products production and sales), the people’s daily life (e.g., supermarket stores, food production and supply and other industries) and other enterprises other than those related to important national economy and people’s livelihood are all shut down. PM_2.5_ exposure was enhanced in April 2020 compared to March 2020, particularly in Hunan, Hubei, Jiangxi, Yunnan and the Pearl River Delta, due to the fact that with the vast majority of the country’s enterprises’ after more than a month of adjustment, most companies have regained their original capacity, accompanied by the population movement caused by the resumption of production and the enhanced plant emissions led to an uptick in PM_2.5_ exposure in April 2020.

### 3.3. Health Risk Assessment of PM_2.5_ during COVID-19 Outbreaks

The results are based on the health risk assessment of PM_2.5_, and the potential risks of disease from PM_2.5_ shown in the study results all exclude diseases due to the COVID-19 epidemic. The results of PM_2.5_ health risks in China (Figure 8) show that the maximum potential death toll in December 2019 was 139,424, and the areas with high PM_2.5_ health risks cover the largest area. In March 2020 and April 2020, the impact of PM_2.5_ was lower than that in December 2019 and January 2020. In April 2020, areas with high health risks covered the lowest area, and the maximum potential death toll was 33,462. Compared with the decrease of 105,962 people in December 2019, the decline was relatively large. In terms of geographical distribution, the health risks of PM_2.5_ in the western region are significantly lower than those in the central and eastern regions. The areas with higher PM_2.5_ health risks are mainly concentrated in Beijing, Tianjin, Hebei, Henan, Chengdu, Chongqing, Pearl River Delta, northwestern XAR and Shanghai. According to the population density map based on LBS data (Figure 5), the population during the Spring Festival transport in 2020 will migrate from first-tier cities and new first-tier cities to small cities. According to data from the Beijing Railway Administration, during the Spring Festival travel period, Beijing sent a total of 8,274,100 passengers, and the population density in Beijing decreased. Population migration has reduced the population density of first-tier cities and new first-tier cities, reducing PM_2.5_ exposure and reducing the health risks caused by PM_2.5_. With the evolution of the COVID-19 epidemic, communities across the country were blocked between the end of January 2020 and March 2020, traffic on the roads was reduced, airports, railway stations and other places of passenger access were strictly controlled, and population movement was reduced, resulting in a reduced health risk from PM_2.5_.

The health risk caused by PM_2.5_ was low in the Tibet Autonomous Region, Qinghai, western Sichuan, northwestern Gansu, and central XAR. The COVID-19 epidemic did not have a significant impact on the health risk of PM_2.5_. From the topography (Figure 1a) and population density map (Figure 5) of the study area, it can be found that these areas have complex topography and a large degree of undulation, which are not suitable for human habitation, making the population density lower and the health risk caused by lower PM_2.5_ concentrations in these areas smaller. In addition, the GDP per capita (Figure 1c) in these regions is low, and the output value of the secondary industry (Figure 1d) is also low, making economic development relatively backward. Moreover, these regions have primary agriculture, animal husbandry, fishery, and tertiary tourism as the pillar industries of their economies, and there is little air pollution caused by PM_2.5_ from large industries, so the health risk caused by PM_2.5_ is relatively low.

### 3.4. Driving Factors Affecting PM_2.5_ Concentration

PM_2.5_ poses a threat to human health, in order to explore the impact of PM_2.5_ caused by anthropogenic activities and aids the reduction of health risk, the study selected six driving factors (i.e., GDP, road area, green area, number of taxis, number of buses, and number of factories) from POI data and used the GWR method to quantify the impact of these variables on PM_2.5_. In order to effectively explore the primary driving factors on PM_2.5_ concentration, this study respectively selects the adaptive Gaussian Akaike information criterion (AIC) index (determine the best bandwidth through the minimum information criterion) to be the kernel function and bandwidth of the GWR model, the correlation between the above driving factors and PM_2.5_ concentration was analyzed.

The difference in spatial regression coefficients between PM_2.5_ and various variables reveals the spatial heterogeneity of PM_2.5_ concentration. The spatial regression coefficient of the GWR model reflects the degree of influence of each variable on the PM_2.5_ concentration (Figure 9). When the spatial regression coefficient is positive, it means that the variable is positively correlated with the PM_2.5_ concentration, and the larger the variable is, the greater the impact on PM_2.5_, and vice versa. From the parameter estimation and validation results of the GWR model (Table 1), coefficients of determination ($R^{2}$) of the six explanatory variables are above 60%, indicating that the GWR model can better fit the relationship between PM_2.5_ concentration and each variable.

From Figure 9a, the national PM_2.5_ concentration and GDP show a negative correlation (Table 2), indicating that the GDP is high and the corresponding PM_2.5_ concentration is low. The most obvious performance is the urban agglomeration in the middle and lower reaches of the Yangtze River, with the largest absolute value of the regression coefficient. The higher the total urban GDP is, the more attention is paid to the reduction of PM_2.5_ pollution. Regression coefficients ranged from −10 to −5 × 10^−3^ in areas such as Ningxia Hui Autonomous Region and central Inner Mongolia, where industry is underdeveloped (Figure 1d), the economy is mainly based on primary and tertiary services, and urban PM_2.5_ emissions are low, so GDP and PM_2.5_ show a strong negative correlation. In the southern part of Guangxi Zhuang Autonomous Region and the three provinces of Northeast China, with correlation coefficients between 0 and 5, as the output value of secondary industries in these regions was larger (Figure 1d), and PM_2.5_ emissions from factories were higher, so GDP showed a weak positive correlation with PM_2.5_.

As shown in Figure 9b, urban green areas in South China, East China, Central China, North China, Chengdu-Chongqing urban agglomeration showed negative correlation with PM_2.5_, with correlation coefficients between −5 × 10^−3^ and 0, indicating that the larger the urban green area, the lower the urban PM_2.5_ concentration. Green plants can not only absorb air pollutants such as dust, sulfide, nitrous oxide, etc., but also the coronal layer of the plant will reduce the speed of wind and block particles in the air. Plant leaves will absorb dust during respiration and photosynthesis. Official data assumes that in an acre of forest planted with 100 trees, 22−60 tons of dust can be absorbed in a year [53]. As a result, urban greenery has a stronger cleaning and air purification function. In the southwest and the west, except for most of the Chengdu-Chongqing urban agglomeration and the northwest, there is a weak positive correlation between the urban green area and PM_2.5_, and the correlation coefficient is between 0 and 5 × 10^−3^. Since these areas are located in the first and second terrain (Figure 1a), the climatic conditions and precipitation conditions are not conducive to large-scale vegetation growth. Most vegetation types are grasslands and low shrubs, and the absorption effect of PM_2.5_ is significantly weak, while the absorption intensity of PM_2.5_ by tall vegetation are intensive in the eastern region [54].

As shown in Figure 9c–e, road area and the numbers of urban taxis and buses show a positive correlation with PM_2.5_ concentration (Table 2). The average regression coefficient between road area and PM_2.5_ is between 0 and 5 × 10^−3^ The population density of Inner Mongolia Autonomous Region, western Yunnan and Tibet Autonomous Region is low due to topography (Figure 1a), climate, and other factors (Figure 5). The roads are scattered, giving the road area and PM_2.5_ a positive correlation. The regression coefficients for the number of urban taxis and PM_2.5_ ranged from −5 × 10^−3^ to 0, with a weak negative correlation in Northeast China, where industry contributed the most to PM_2.5_ due to the concentrated distribution of heavy industry in the region (Figure 1d). The regression coefficient between the number of urban buses and PM_2.5_ is from 0 to 5 × 10^−3^ and the Jing-Jin-Ji urban agglomeration has a strong positive correlation with Shanxi and eastern Inner Mongolia, with the regression coefficient from 5 × 10^−3^ to 10 × 10^−3^. Road area is proportional to urban traffic emissions, with higher emissions (PM_2.5_, nitrogen oxides, carbon oxides, etc.) from urban traffic with major road. According to the China Mobile Source Environmental Management Annual Report (2019) released by the Ministry of Ecology and Environment, China has been the world’s largest producer and seller of motor vehicles for 10 consecutive years and, in 2018, the total amount of four pollutants emitted from motor vehicles in the country was initially accounted for 40.653 million tons. This includes 442,000 tons of particulate matter. Automobiles are the main contributors to air pollution emissions from motor vehicles [55]. The topography of Yunnan, Guizhou, and southwestern Sichuan (Figure 1a) resulted in a weak negative correlation between the number of bus and PM_2.5_.

From Figure 9f there is a significant positive correlation between the number of factories in Central, North, Southwest and Northwest China on PM_2.5_, with regression coefficients between 0 and 20 × 10^−3^, with the strongest correlation in Shanxi and Central Inner Mongolia. The distribution of secondary industry values in China (Figure 1d) shows that these regions have higher secondary industry emissions. The greater the number of urban factories, the higher the emissions of pollutants (nitrogen oxides, carbon oxides, PM_2.5_, etc.) resulting from the factories’ production activities. The spatial regression coefficients of the number of factories and PM_2.5_ show an increasing trend from east to west of China, while the South China show a weak negative correlation trend, indicating that South China pays attention to the emission of PM_2.5_ from factories. Meanwhile, South China is close to the ocean, and the PM_2.5_ concentration generated by factories is positively influenced by the sea breeze that accompanies the ocean currents [56].

The quantitative analysis of the driving factors of PM_2.5_ found that the number of urban factories has the greatest impact on PM_2.5_, indicating that industrial emissions dominated the China’s PM_2.5_ pollution. In terms of reducing the health risks and pollution prevention measures caused by PM_2.5_, industrial emissions should be controlled. Launching laws and regulations for PM_2.5_ prevention and control, and formulating relevant standards for industrial waste gas and waste emissions from factories, is necessary. The relevant departments should ensure the implementation of the system. The government promotes the location of new factory sites, increases planning for PM_2.5_ prevention and control in industrial parks, and provides technical support for replacing new clean equipment in factories, while the authorities will increase the supply of clean energy such as electricity and natural gas to reduce reliance on traditional energy sources in factories and reduce PM_2.5_ emissions. Vegetation coverage is an important part of urban ecological construction. Relevant planning departments should make reasonable plans for vegetation planting areas around factories and inside cities to prevent the spread of PM_2.5_ from further harming surrounding residents’ health, because trees play a unique ecological function in the purification of atmospheric fine particles.

## 4. Conclusions

In this paper, we have evaluated the population exposure and health risk of PM_2.5_ in the context of the COVID-19 epidemic, as well as spatiotemporal distribution and driving factors of PM_2.5_ from spatiotemporal big data, including station monitoring of PM_2.5_ concentration, urban POI, and LBS data. The EOF study of the national monthly PM_2.5_ distribution pattern and its evolution in 2014–2019 found that, the time series coefficients of the first feature vector of monthly PM_2.5_ in 2014–2019 showed obvious characteristics, and the time series coefficients showed a downward and then upward trend from year to year, roughly in a U-shaped distribution. In terms of spatial distribution, PM_2.5_ concentrations in central and northern China are relatively higher than those in the surrounding areas, and the distribution pattern shows attenuation from central and northern China to its surrounding areas. From 2014 onwards, PM_2.5_ gradually changed from one center (Central China and North China) to two centers (Central China, North China and XAR). The population density distribution based on the LBS data in China shows that the eastern areas are concentrated and the western areas are relatively sparse. In March and April 2020, most of the PM_2.5_ concentration were below $35\;\mu g/m^{3}$, which is caused by the control measures on population movement and industrial production in the cities during the COVID-19 epidemic. The health risk of PM_2.5_ in the western region is significantly lower than that in central and eastern regions. After the outbreak of COVID-19, the domestic health risk caused by PM_2.5_ significantly reduced. The GWR of PM_2.5_ with GDP, urban green space, road area, number of urban taxis, number of urban buses and number of urban factories demonstrates that GDP and urban green space were negatively correlated with PM_2.5_, while road area, number of urban taxis, number of urban buses and number of urban factories were positively correlated with PM_2.5_. In terms of reducing the health risks and pollution prevention measures brought about by PM_2.5_, it is recommended that relevant authorities restrict factory emissions and promote new energy transportation and encourage residents to travel green.

## Figures and Tables

**Figure 1 ijerph-17-07664-f001:**
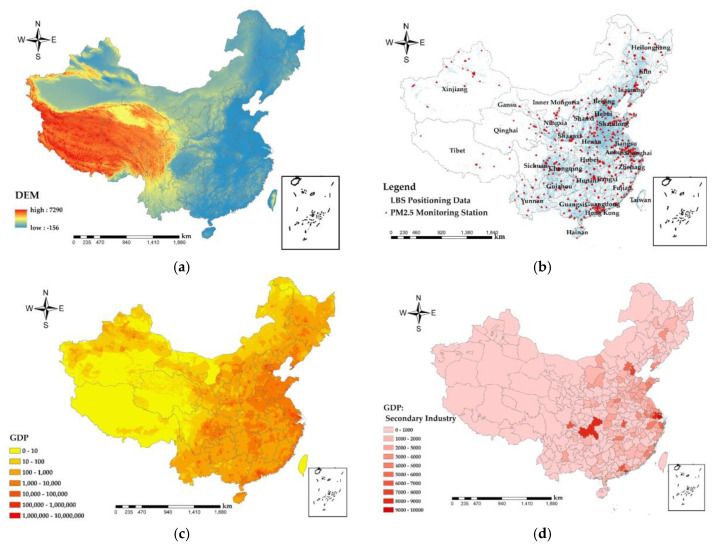
Study area and data, (**a**) elevation, (**b**) spatial distribution of particulate matter (PM_2.5_) monitoring sites and Tencent’s location-based services (LBS) in China, (**c**) gross domestic product (GDP), and (**d**) secondary production.

**Figure 2 ijerph-17-07664-f002:**
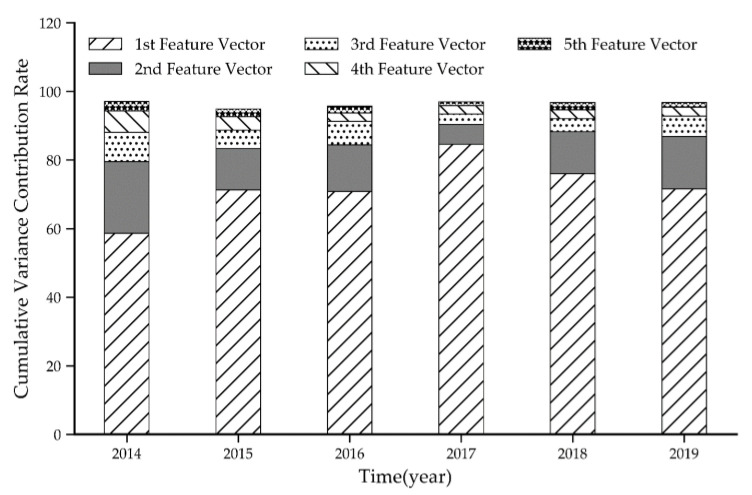
Variance contribution rate of each feature vector.

**Figure 3 ijerph-17-07664-f003:**
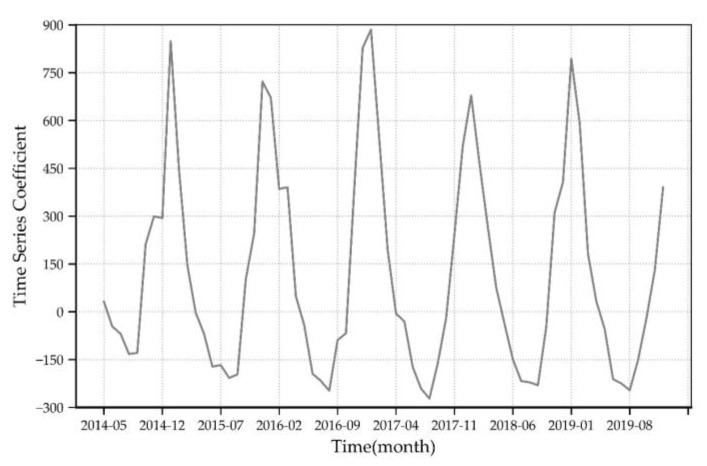
Monthly feature vector time series coefficients during 2014–2019.

**Figure 4 ijerph-17-07664-f004:**
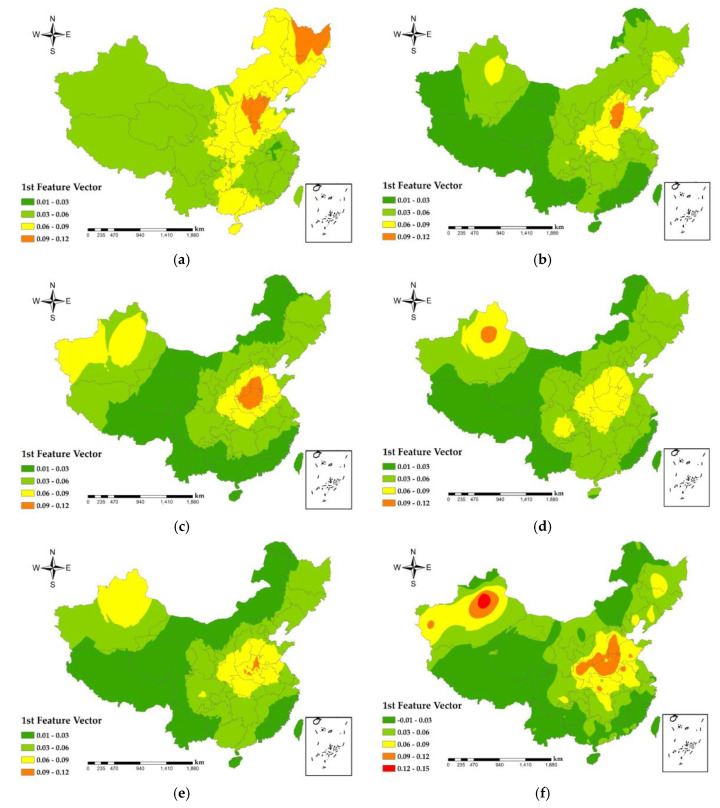
Spatial distribution of the first feature vector of monthly PM_2.5_ concentration during 2014–2019. (**a**) 2014; (**b**) 2015; (**c**) 2016; (**d**) 2017; (**e**) 2018; and (**f**) 2019.

**Figure 5 ijerph-17-07664-f005:**
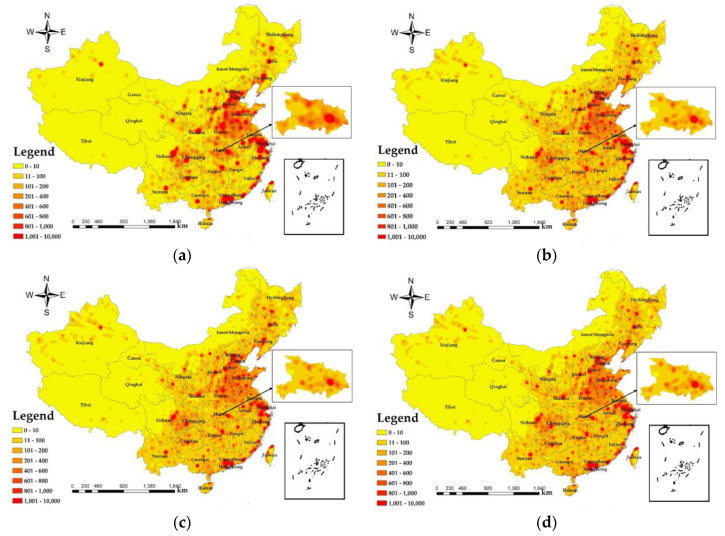
Population density distribution map based on LBS data. (**a**) 25 December 2019; (**b**) 11 January 2020; (**c**) 7 March 2020; and (**d**) 3 April 2020.

**Figure 6 ijerph-17-07664-f006:**
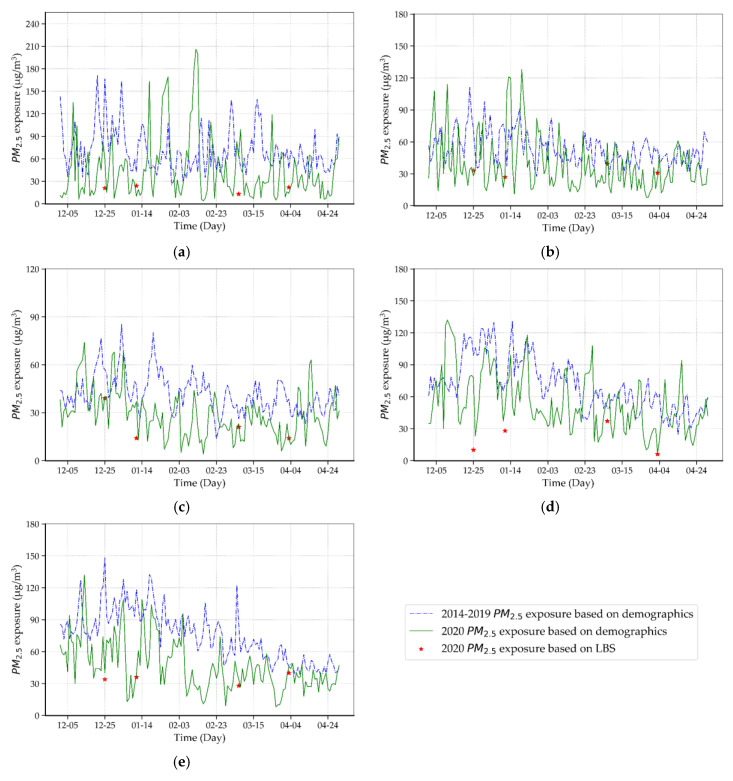
PM_2.5_ exposure in typical population-intensive cities based on demographic and LBS data. (**a**) Beijing, (**b**) Shanghai, (**c**) Guangzhou, (**d**) Chengdu, and (**e**) Wuhan.

**Figure 7 ijerph-17-07664-f007:**
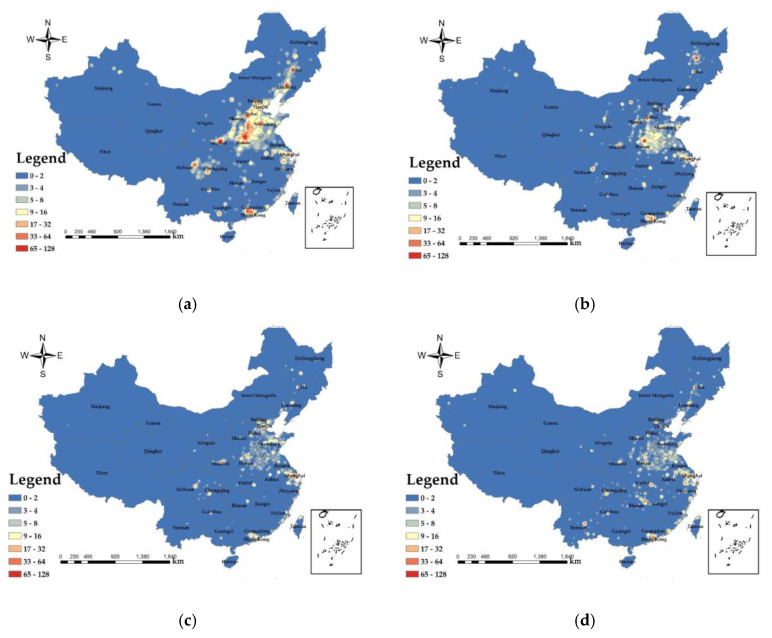
PM_2.5_ exposure assessment value (**a**) 25 December 2019; (**b**) 11 January 2020; (**c**) 7 March 2020; and (**d**) 3 April 2020.

**Figure 8 ijerph-17-07664-f008:**
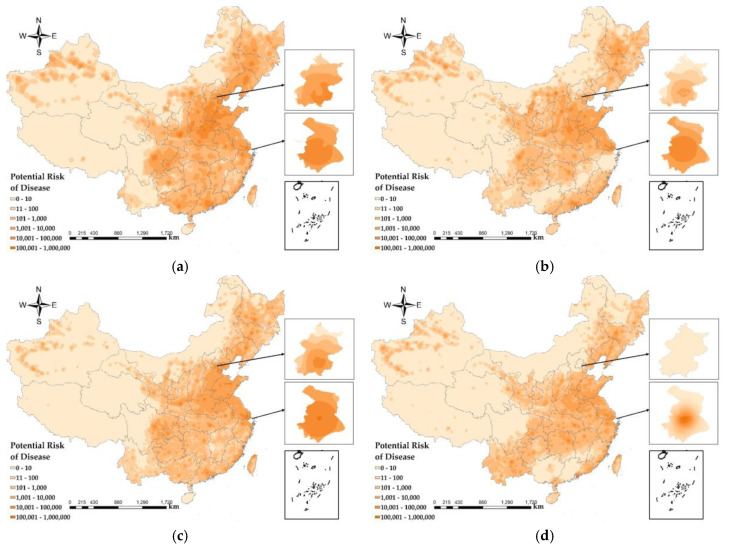
PM_2.5_ health risks assessment. (**a**) 25 December 2019; (**b**) 11 January 2020; (**c**) 7 March 2020; and (**d**) 3 April 2020.

**Figure 9 ijerph-17-07664-f009:**
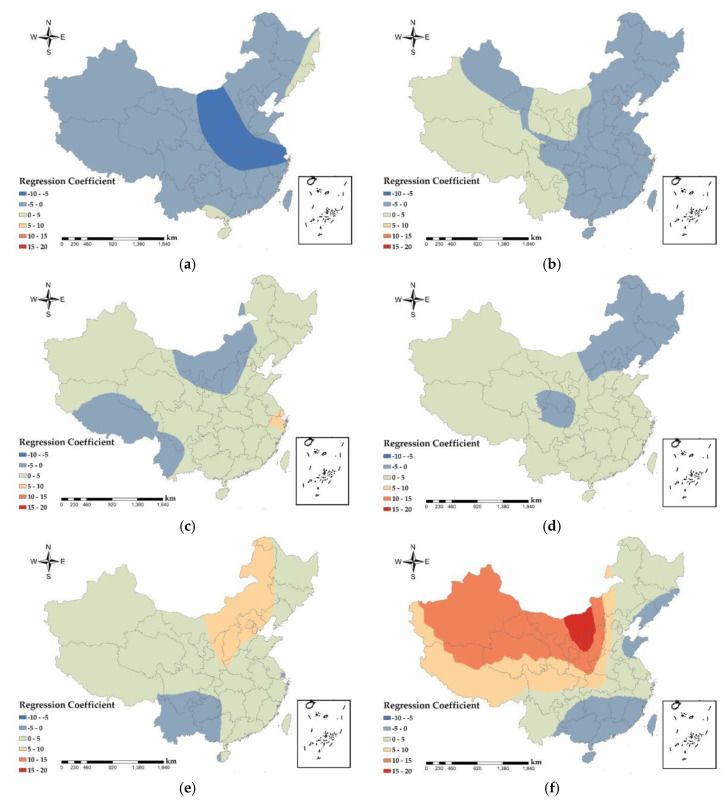
Spatial distribution of regression coefficient (×$10^{-3}$) between PM_2.5_ concentration and six explanatory variables. (**a**) urban GDP; (**b**) the area of urban green space; (**c**) road area; (**d**) number of taxis; (**e**) number of buses; and (**f**) number of factories.

**Table 1 ijerph-17-07664-t001:** Estimation and validation of regression results of the geographically weighted regression (GWR) model.

Year	AICc	$R^{2}$	$\mathrm{Adjusted}\;R^{2}$	Residual Squares
2014	2094.78	0.6759	0.6328	19078.16
2015	2095.02	0.6910	0.6501	19473.78
2016	2107.93	0.6303	0.5869	20026.23
2017	1977.24	0.6738	0.6315	13311.88

**Table 2 ijerph-17-07664-t002:** Mean value of spatial regression coefficient of GWR model (×10^−3^).

Year	Urban GDP	Road Area	Green Space Area	Taxis	Buses	Factories
2014	−6.3	2.1	−0.8	1.7	1.9	7.4
2015	−4.8	1.7	−0.5	1.1	2.3	3.3
2016	−0.5	1.4	−0.4	0.8	2.8	3.1
2017	−1.7	1.1	−0.4	0.4	1.9	0.8

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
