# Peer review of "Spatiotemporal Big Data for PM2.5 Exposure and Health Risk Assessment during COVID-19"

_ijerph, 2020, doi:10.3390/ijerph17207664_

Round 1

Reviewer 1 Report

Review of the manuscript “Spatiotemporal Big Data for PM2.5 Exposure and Health Risk Assessment during COVID-19

General comments

In general the paper consists of two parts. In the first part the authors analyse the relationship between the place of interest (POI) data (e.g. urban green space, road transport) and PM2.5 concentrations in China. In the second part the impact of COVID-19 on PM2.5 population exposure and health effects are studied. The proplem of this study is that the connection between the first and the second part is not well defined. It is not clearly written how the first analysis can help in understanding the COVID-19 part. Additionally, the exposure analysiss was done for four selected dates. I have not found the expalantion why the authors choose these dates. The results for individual dates can be under a great impact of specific meteorological conditions.
The authors describe in details the results that we can see in the plots instead of deeper analysis and conclusions.

Other comments:

Section 2.1

  • provide more information about the POI data. Is it grid data? What is the spatial resolution? This is important as you use “Big data” in the title but not sufficiently explain these data.
  • Provide more information about the LBS data. How many people from the total population is included in this data?

Line 138 “it is difficult to obtain better results” – better than what?

Line 147 “the accuracy of the results is higher” – higher than what?

Line 165 – explain why 10 ug/m3. Did you calculate it or assume it?

Figure 5 – you should use the same scale for all figs (a-f). It will enable to compare the plots visually.

Line 187 “that the annual PM2.5 pollution is most severe in winter and spring.” – I think monthly or seasonsal – not annual. Second, this is probably well known that the lowest air pollution is in the winter season?

Lines 203 – 228 – you describe in details the main reasons of high concentrations in some areas. Is it the direct result of your study? It is not clear for me if you use your datasets to it?

Table 2 – you have not introduces the spatial regression coefficient in the methods section. What is the range of this coefficient?

Figure 6 – It is not clear why you calculate the density for four selected dates. Why these dates?

342-375  – The results for few selected dates can be under influence of specific meteorological conditions, so you can not draw conclusions on the general situation in the month of interest.

414-440 – this is quite general and not strictly related to your results.

Author Response

Thanks for the constructive suggestions. We revised comments item by item. Please refer to the attachment for details

Reviewer 2 Report

This study analysed the PM2.5 exposure and health risk assessment during COVID-19. Quantitative assessment for PM2.5 and it’s health risk are assessed.  The methodology sections are clearly presented and the novelty of this study is enough to be accepted for publication. Following suggestions could improve the overall quality of the present article;

  1. Authors need to improve the abstract section. The significance of the study should highlight in the abstract section.
  2. Introduction section contains enough informations. Authors could cite some recent COVID related papers in the introduction section.
  3. Authors need to explain equation 7 a bit more. It is some exiting equations or authors developed this analytical equation?
  4. Line 165, authors claim the mortality rate decreased by 6%. Is it findings of this study? If no, authors need to cite the corresponding article.
  5. X and Y axis legend of the figure 2 and 3 should improve. It’s not visible.
  6. Same comment for all figures.

Author Response

Q1: Authors need to improve the abstract section. The significance of the study should highlight in the abstract section.

A1: We are grateful about your kind suggestions. We have rewritten the abstract. First, the relevant status of COVID-19 is mentioned, and the significance of this study is clarified. Secondly, we briefly introduced the propose of the study, and then, the detailed methods and data are presented. Thirdly, some results and relative solution are stated. Finally, the conclusion of this study is summarized.

Q2: Introduction section contains enough information. Authors could cite some recent COVID related papers in the introduction section.

A2: We are so appreciated with your suggestion. We have adjusted the order of background introduction, and cite some references related to COVID-19 to make the introduction section more complete.

Guan W, Ni Z, Hu Y, et al. Clinical characteristics of coronavirus disease 2019 in China. New England journal of medicine, 2020, 382(18): 1708-1720.

Zhou F , Yu T , Du R , et al. Clinical course and risk factors for mortality of adult inpatients with COVID-19 in Wuhan, China: a retrospective cohort study. The Lancet, 2020, 395(10229).

Q3: Authors need to explain equation 7 a bit more. It is some exiting equations or authors developed this analytical equation?

A3: Thank you for your suggestions. We cite a reference of our previous work, which is the prototype of Equation 7 in this study. Equation aims to estimate the population-weighted PM2.5 exposure from LBS data, which can effectively exploit the advantages of high spatial and temporal resolution of data.

Shen Y, Yao L. PM2. 5, population exposure and economic effects in urban agglomerations of China using ground-based monitoring data[J]. International journal of environmental research and public health, 2017, 14(7): 716.

Q4: Line 165, authors claim the mortality rate decreased by 6%. Is it findings of this study? If no, authors need to cite the corresponding article.

A4: This part has been amended, and relevant references are cited in paragraph 1, section 2.4. The literature describes the health risks associated with PM2.5 as follows: when PM2.5 concentration is reduced from  to , exposure reduces the risk of death by approximately 6%.

World Health Organization. WHO Air quality guidelines for particulate matter, ozone, nitrogen dioxide, and sulfur dioxide: global update 2005: summary of risk assessment. World Health Organization, 2006.

Q5: X and Y axis legend of the figure 2 and 3 should improve. It’s not visible.

Same comment for all figures.

A5: Thank you for your constructive suggestion. We have modified legend and scale to make the figure 2 and 3 more readable. Moreover, we checked all the figures in the manuscript, and made improvements.

Reviewer 3 Report

The manuscript used location-based service (LBS) data and monitoring data to analyze the health risk during the COVID 19 pandemic.

The manuscript is organized well and the conclusion is demonstrated clearly. Minor revision is recommended.

Almost labels in all figures are invisible.

Please change to high-resolution figures and enlarge the font. In addition, please change "PM2.5" to "PM2.5" to keep consistent.

Author Response

Q1: Almost labels in all figures are invisible.

A1: Thank you for your constructive suggestion. We checked all the figures in the manuscript, and made improvements.

Q2: Please change to high-resolution figures and enlarge the font. In addition, please change "PM2.5" to "PM2.5" to keep consistent.

A2: We are grateful for your kind suggestion. We have redrawn the figures to ensure each figure more readable, including the resolution, legend, font, etc. In addition, all the incorrect expressions of “PM2.5” have been modified to “PM2.5”.

Round 2

Reviewer 1 Report

I like the idea of application of LBS data and GIS methods to calculate population exposure and health risk. Spatiotemporal analysis of PM2.5 concentrations is also interesting. I can see that the manuscript has been significantly improved in terms of description od input data and in general data analysis. I am really sorry but I am not convinced that analysis for 4 dates can show the influence of covid-19 situation on air pollution. The authors have included information on rainfall and wind speed for these dates. This is fine. However, they also emphasize in the text that air pollution concentrations are highly dependant on temperaure. Information on temperature has not been include for these selected dates; we should also remember that concetrations of PM will also depend on e.g. day of week (weekday or weekend). To sum up – the analysis is interesting but in my opinion you can not draw conclusions on impact of covid situation on air pollution based on your analysis.

Author Response

Thanks for the constructive suggestions. We revised them, and please refer to the attachment for details.
